# Blood Biomarkers in Takotsubo Syndrome Point to an Emerging Role for Inflammaging in Endothelial Pathophysiology

**DOI:** 10.3390/biom13060995

**Published:** 2023-06-15

**Authors:** Michiaki Nagai, Sergey Shityakov, Manuel Smetak, Hannah Jill Hunkler, Christian Bär, Nicolas Schlegel, Thomas Thum, Carola Yvette Förster

**Affiliations:** 1Department of Cardiology, 2-1-1, Kabeminami, Aaskita-ku, Hiroshima City Asa, Hiroshima 731-0293, Japan; 2Infochemistry Scientific Center, Laboratory of Chemoinformatics, ITMO University, Lomonosova Str. 9, 191002 Saint-Petersburg, Russia; shityakoff@hotmail.com; 3Department of Anaesthesiology, Intensive Care, Emergency and Pain Medicine, University of Würzburg, 97080 Würzburg, Germany; manuelsmetak@gmail.com; 4Institute of Molecular and Translational Therapeutic Strategies (IMTTS), Hannover Medical School, 30625 Hannover, Germany; hunkler.hannah@mh-hannover.de (H.J.H.); baer.christian@mh-hannover.de (C.B.); thum.thomas@mh-hannover.de (T.T.); 5REBIRTH-Centre for Translational Regenerative Medicine, Hannover Medical School, 30625 Hannover, Germany; 6Fraunhofer Institute for Toxicology and Experimental Medicine (ITEM), 30625 Hannover, Germany; 7Department of General, Visceral, Transplant, Vascular and Pediatric Surgery, University of Würzburg, 97080 Würzburg, Germany; schlegel_n@ukw.de

**Keywords:** Takotsubo syndrome, catecholamines, inflammation, aging, inflammaging, glucocorticoid receptor, cytokine, epinephrine, norepinephrine, dopamine

## Abstract

Takotsubo syndrome (TTS), an acute cardiac condition characterized by transient wall motion abnormalities mostly of the left ventricle, results in difficulties in diagnosing patients. We set out to present a detailed blood analysis of TTS patients analyzing novel markers to understand the development of TTS. Significant differences in proinflammatory cytokine expression patterns and sex steroid and glucocorticoid receptor (GR) expression levels were observed in the TTS patient collected. Remarkably, the measured catecholamine serum concentrations determined from TTS patient blood could be shown to be two orders of magnitude lower than the levels determined from experimentally induced TTS in laboratory animals. Consequently, the exposure of endothelial cells and cardiomyocytes in vitro to such catecholamine concentrations did not damage the cellular integrity or function of either endothelial cells forming the blood–brain barrier, endothelial cells derived from myocardium, or cardiomyocytes in vitro. Computational analysis was able to link the identified blood markers, specifically, the proinflammatory cytokines and glucocorticoid receptor GR to microRNA (miR) relevant in the ontogeny of TTS (miR-15) and inflammation (miR-21, miR-146a), respectively. Amongst the well-described risk factors of TTS (older age, female sex), inflammaging-related pathways were identified to add to these relevant risk factors or prediagnostic markers of TTS.

## 1. Introduction

Takotsubo syndrome (TTS), also known as broken heart syndrome, stress-induced cardiomyopathy, and Takotsubo cardiomyopathy, is an acute cardiac syndrome with rapid onset of chest pain and dyspnea [1,2]. TTS is often triggered by physical and/or emotional stress and is characterized by a transient and reversible severe left ventricular dysfunction which typically recovers spontaneously within hours to weeks [3]. TTS is remarkably similar to acute coronary syndrome (ACS), with almost the same clinical presentations and ST elevations; therefore, differential diagnosis is critically important in the emergency department. The prevalence of TTS has been reported to be approximately 2–3% of all patients with clinical appearance of ACS [3,4].

At present, TTS-related heart failure is believed to be chiefly induced by a so-called catecholamine (CAT) storm, e.g., increased circulating and myocardial CAT levels with myocardial toxicity [5,6,7,8]. Central to this unifying hypothesis, the abnormal left ventricular contraction pattern is attributed to this CAT surge which triggers left ventricular wall motion abnormality, e.g., widespread dyskinesia in the apical segments and hyperkinesia in the basal segment of left ventricle (LV) with apical ballooning [3,4]; available information is, however, controversial [9,10,11]. As a pathophysiological mechanism, it is suggested that local differences in left ventricular (LV) adrenergic receptors may underly this LV wall motion abnormality [12,13]. In fact, experimental studies showed that the LV in canines has β2-adrenoceptors (β2-AR) that are expressed much more in the apical than in the basal segments [14]; the signaling pathways leading to this aetiopathogenesis, at any rate, remain unclear. Feola et al. further supported this hypothesis with a myocardial PET study showing decreased coronary flow reserve and impaired metabolism in the apical segments during the acute phase of TTS [15].

Additional possible pathophysiological mechanisms have been suggested, such as induced switch in signal trafficking and autonomic nervous system dysfunction with cardiac sympathetic activation, including the mentioned overstimulation of beta receptors [13,16,17,18,19]. Female sex and age-related lack of female sex hormones are further well-known characteristics of TTS patients [20].

Given that TTS is remarkably similar to ACS, with almost the same clinical presentation and ST elevations, to date, in the clinical practice, it has been difficult to differentiate TTS from ACS at the at the moment of onset. No ECG criteria have been identified that reliably discriminate between TTS and ST-segment elevation myocardial infarction [21]. The ECG changes are transient and resolve within months in most cases.

Recent studies have identified microRNAs (miRNAs/miRs) as potentially promising sensitive and specific biomarkers for cardiovascular diseases such as TTS, including a unique signature of miR-1, miR-16, miR-26a, and miR-133a, which differentiates TTS patients from STEMI patients [22,23].

Nef et al. examined endomyocardial biopsy samples from 16 TTS patients and observed the increased activation of the PI3K and Akt pathways, which were later found to return to normal levels during follow-up and were not present in cases of ischemia [24]. These pathways are associated with β2AR-Gαi activity and play a role in promoting the survival of cardiomyocytes. Furthermore, Nakano et al. found elevated levels of proteins involved in β2AR-Gαi signaling, including GRK and β-arrestin, in tissue samples from 26 patients with acute TTS, highlighting their importance in facilitating β2AR-Gαi stimulus trafficking [25].

A study comparing TTS with myocardial infarction (MI) found that the former pathology primarily affects women and has distinct demographic and comorbid predictors, including a higher prevalence of cerebrovascular accidents, drug abuse, anxiety disorders, mood disorders, malignancy, chronic liver disease, and sepsis. These findings highlight the importance of considering these factors when diagnosing TTS and differentiating it from MI [26]. In a multicenter study, patients with TTS showed increased inflammation markers, including ultrasmall superparamagnetic particles of iron oxide enhancement in the myocardium, elevated serum interleukin-6 concentrations, and altered blood monocyte subpopulations compared to control subjects. However, these inflammatory changes were resolved at 5 months follow-up, except for persistent elevations in serum interleukin-6 concentrations and reductions in intermediate monocyte subpopulations [27].

Using blood samples from real-life patient cohorts, we set out to comprehensively quantify and characterize CAT levels and other blood-borne biomarkers (markers of inflammation, physical/emotional stress hormone, and endogenous sex hormone levels and their receptors) to identify a set of novel TTS-associated biomarkers, pointing to inflammaging-associated triggers for TTS rather than direct CAT effects on cardiomyocytes and endothelial cells of the cardiovascular and cerebrovascular segments. By the help of systems biology, we delineate new targets and specific molecular pathways which will allow the development of precision medicine in the prevention and therapy of TTS in the future.

## 2. Materials and Methods

### 2.1. Study Population and Human Patient Serum

Patient serum preparation from human blood was performed as described previously [28,29]. Briefly, human blood was collected using S-Monovette tubes (Sarstedt). We sampled blood from 25 pseudonymized patients and subsequently divided into 5 (6 for cytokine measurements) different groups: postmenopausal healthy patients, *n* = 3, premenopausal healthy patients, *n* = 3, ACS acute (<6 weeks), *n* = 4, TTS acute (first 2 weeks after onset), *n* = 4, TTS subacute (2–6 weeks after onset), *n* = 8, and TTS chronic (>6 weeks), *n* = 3. All patients were diagnosed according to the InterTAK Diagnostic Criteria [30]. Healthy control subjects were selected that did not present with altered coronary arteries in the coronary angiography diagnostics, while minor cardiovascular or endocrinological diseases were accepted (e.g., arterial hypertension, diabetes mellitus). 

For serum preparation, all the blood was drawn from subjects in using S-Monovette collection tubes (Sarstedt) and incubated at room temperature for 60 min. Then, the blood samples were centrifuged at 1500× *g* for 15 min, and the serum was isolated, immediately frozen, and stored at −80 °C until the extraction. All the methods used in this study were performed in accordance with the relevant guidelines and regulations. Serum samples were quantified and analyzed individually.

### 2.2. Ethics Approval and Consent to Participate

The study protocol was approved by the Hiroshima City Asa Hospital Research Committee (serial number 01-3-3), Hiroshima, Japan, as previously described [28], and was conducted in accordance with the principles stated in the Declaration of Helsinki. All participants provided informed written consent. 

Briefly, as published previously [28,29], from July 2019 to September 2019, hospitalized TTS patients were registered consecutively at Hiroshima City Asa Hospital. All patients were diagnosed according to the InterTAK Diagnostic Criteria [30]. All of the 25 subjects included in this analysis had good prognosis. None of the patients died during hospitalization. In the Kruskall–Wallis test, there were no significant differences in age (median age 83 vs. 80 years, *p* = 0.48). In total, ¾ of the TTS patients were of female sex. 

For control purposes, blood from 3 healthy premenopausal females and from 3 healthy males (age 25–50 years) was collected.

### 2.3. Study Design—Human Patient Serum

For serum preparation, all the blood was drawn from subjects in using S-Monovette collection tubes (Sarstedt) and incubated at room temperature for 60 min [24]. Then, the blood samples were centrifuged at 1500× *g* for 15 min, the serum was isolated, immediately frozen, and stored at −80 °C until the extraction. All the methods used in this study were performed in accordance with the relevant guidelines and regulations. Serum samples were quantified and analyzed individually.

### 2.4. Biochemical Analyses

Blood samples were drawn after 10 min rest in the supine position in the morning. Adrenaline, noradrenaline, and dopamine concentrations in the plasma were assessed by high-performance liquid chromatography (SRL Inc., Tokyo, Japan). Serum cortisol concentrations were determined by electrochemiluminescence immunoassay (Elecsys^®^, Roche Diagnostics, Basel, Switzerland). While testosterone was measured using radioimmunoassay kits, 17β-estradiol was measured using an ultra-sensitive radioimmunoassay kit (Elecsys^®^, Roche Diagnostics, Basel, Switzerland).

### 2.5. Endothelial Cell Cultures

The mouse brain capillary endothelial cell line cEND and MyEND were immortalized, isolated as described previously [31,32,33], and cultured in Dulbecco’s modified Eagle’s medium (DMEM) with high glucose (91625 Schnelldorf Sigma-Aldrich) supplemented with l-glutamine, MEM-vitamin solution, non-essential amino acids (NEA), Na pyruvate, penicillin/streptomycin (P/S) (all from Sigma), and 10% heat-inactivated fetal calf serum (FCS) [34].

The human brain vascular endothelial cell line hCMEC/D3 [35,36] was cultivated in Microvascular Endothelial Cell Growth Medium Kit Enhanced (PELOBiotech, Planegg, Germany) with all supplements (FCS, Glutamin, EGF, b-FGF, VEGF, R3-IGF-1, Hydrocortison, Gentamycin) added. 

For the different treatment regimens, cEND and hCMEC/D3 cells were grown to confluence for 5 days using Transwell systems. For this, the cells were placed on 6-well or 12-well Nunc Polycarbonate Cell Culture Inserts (Thermo Fisher Scientific, #140640 or #140652) with pore sizes of 0.4 µm coated with gelatine. Dulbecco’s Modified Eagle’s Medium (DMEM)—high glucose (91625 Schnelldorf Sigma-Aldrich, #D5796) with the addition of 10% heat-inactivated fetal calf serum (FCS), 2% sodium pyruvate, 2% L-glutamine, 2% MEM-vitamins, 2% non-essential amino acid solution, and 50 U/mL penicillin–streptomycin—was used as growth medium. The medium was placed in the lower and upper compartment of the Transwell system. After growing to confluence, differentiation was induced for 24 h using DMEM—high glucose with 1% FCS and 50 U/mL penicillin–streptomycin. The medium in the lower compartment was then removed and serum was added, causing the cells to have contact with differentiation medium from above and serum from below (only divided by the microporous membrane).

### 2.6. Human iPSC Maintenance and Cardiac Differentiation

Human-induced pluripotent stem cells (iPSC) [37] were cultivated in StemMACS medium (StemMACS iPS-Brew XF with supplement, Miltenyi Biotec, #130-104-368) on Geltrex (Gibco, A1413302) and differentiated into cardiomyocytes via Wnt modulation as described before [38,39,40]. Briefly, iPSCs were grown to confluency and passaged with Versene (Gibco, 15040066) in StemMACS medium with 2 µM Thiazovivin (Selleckchem, S1459). Directed cardiomyocyte differentiation was initiated at 70–80% confluency with 5 μM GSK-3 inhibitor XVI (Merck, 361559) in cardio differentiation medium (250 mg human recombinant albumin (Sigma-Aldrich, A9731), 100 mg L-AA (L-ascorbic acid 2-phosphate sesquimagnesium salt hydrate, Sigma-Aldrich, A8960) in 500 mL RPMI medium (RPMI 1640 + GlutaMAX, Gibco, 72400047). for 48 h. Subsequently, medium was changed to cardio differentiation medium supplemented with 5 mM IWP-2 (Selleckchem, S7085), a Wnt signaling inhibitor for 48 h, followed by medium changes with cardio differentiation medium every 48 h. On differentiation day 8, when cells started to contract spontaneously, cardiomyocytes were cultured in 1× B-27 (Gibco, 17504001) in RPMI 1640 + GlutaMAX medium, with medium changes every 2 to 3 days.

Cardiomyocytes were purified by metabolic selection [41] with 4 mM DL-lactate (Merck, L4263, in 1 M HEPES, Carl Roth, HN77.3) in no-glucose RPMI medium (RPMI 1640, no glucose, Gibco, 11879020) with 250 mg human recombinant albumin and 100 mg L-AA. Experiments were performed in iPSC-derived cardiomyocytes between day 55 and 80 of differentiation.

### 2.7. Catecholamine Treatment In Vitro

cEND and MyEND were treated with a physiological and supraphysiological CAT mix—corresponding concentration c1–c3—as determined from TTS patient serum, comp. Table 1, epinephrine (Sigma-Aldrich, E4250), norepinephrine (Sigma-Aldrich, A0937), and dopamine (Sigma-Aldrich, H8502), the respective cell culture media, for 24 h. Epinephrine and norepinephrine were dissolved in 0.5 M HCl, and culture medium was used as a solvent for dopamine.

For CAT treatment of iPSC-derived cardiomyocytes, human iPSC-derived cardiomyocytes were treated with a physiological mix of these CATs, epinephrine (Sigma-Aldrich, E4250), norepinephrine (Sigma-Aldrich, A0937), and dopamine (Sigma-Aldrich, H8502),—corresponding concentration c3/ TTS acute levels as determined from TTS patient serum, comp. Table 1 in RPMI medium supplemented with 1× B-27 (Gibco, 17504001) for 24 h. Additionally to the CAT mix designated c3, cells were treated with a supraphysiological concentration of 5 mM isoprenaline (Sigma-Aldrich, I6504) [42].

### 2.8. Gene Expression Analysis in cEND and MyEND Cells

For the RT-PCR, the cells were washed twice with sterile PBS. The cells were then harvested and lysed, and the RNA was isolated and purified according to the manufacturers’ instructions of Nucleospin RNA (Macherey-Nagel, #740955). A total of 1 µg of the RNA was converted to cDNA using a High-Capacity cDNA Reverse Transcription Kit (Thermo Fisher Scientific, #4368814) in a 2720 Thermal Cycler (Thermo Fisher Scientific, #4359659). The Real-Time qPCR was performed using TaqMan Fast Advanced Mastermix (Thermo Fisher Scientific, #4444557) along with the synthesized cDNA and the following TaqMan Gene Expression Assays (Thermo Fisher Scientific): claudin-5 (Mm00727012_s1, #4331182), occludin (Mm00500912_m1, #4331182), ZO-1 (Mm00493699_m1, #4331182), VE-cadherin (Mm00486938_m1, #4331182), and calnexin as endogenous control (Mm00500330_m1, #4331182). The measurements were performed in a StepOnePlus Real-Time PCR System (Thermo Fisher Scientific, #4376600).

### 2.9. Western Blot Analysis in cEND and MyEND Cells

Cells were washed with ice-cold PBS (twice) and then lysed on ice with ice-cold RIPA buffer containing a protease inhibitor cocktail (ROCHE). After the harvest, the cells were sonicated (10 times for 0.5 s and 20 W) and centrifuged (10 min, 4 °C, 11.000 rcf). The supernatant was saved, and the protein content was quantified with Pierce BCA Protein Assay Kit (Thermo Fisher Scientific, #23225). A total of 20 µg protein (mixed 1:4 with 4 × Laemmli containing 6% β-mercaptoethanol) was separated by electrophoresis using self-made SDS-polyacrylamide electrophoresis gels (separation gel 8%) and subsequently transferred overnight at 4 °C on a PVDF membrane. These membranes were blocked for 1 h with 5% non-fat dry milk in PBS and incubated with the primary antibody (overnight at 4 °C) in 1% BSA in PBS. After washing with 0.1% Tween in PBS (PBS-T, 3 times for 10 min) and blocking with 5% non-fat dry milk in PBS (20 min), the membranes were incubated with the secondary antibody in 1% BSA in PBS (1 h). As primary antibodies we used: mouse anti-claudin-5 (1:1000, Thermo Fisher Scientific, #35-2500), guinea pig anti-occludin (1:100, Acris, #358-504), rabbit anti-ZO-1 (1:1000, Thermo Fisher Scientific, #61-7300), anti-β-actin-HRP (1:25,000, Sigma-Aldrich, #A3854), and anti-goat VE-cadherin (1:200, Santa Cruz Biotechnology, #sc-6458). As secondary antibodies we used: horse anti-mouse IgG (1:3000, Cell Signaling Technology, #7076S), goat anti-guineapig IgG (1:5000, Santa Cruz Biotechnology, #sc2438), goat anti-rabbit IgG HRP-linked (1:3000, Cell Signaling Technology, #7074S), and mouse anti-goat IgG-HRP (1:3000, Santa Cruz Biotechnology, sc-2354). For imaging, the membranes were washed with PBS-T (3 times for 10 min) and incubated in ECL solution (2 min). The images were taken with FluorChem FC2 Multi-imager II (Alpha Innotech). The ImageJ software was used to determine the density of the protein bands.

### 2.10. Gene expression Analysis in Cardiomycytes

RNA were isolated with QIAzol Lysis Reagent (Qiagen, 79306) and reverse transcribed with the Biozym cDNA synthesis kit (Biozym, 3314710X) according to manufacturer’s guidelines. Gene expression levels were measured via RT-qPCR with iQ SYBR Green Supermix (Bio-Rad, 1708880). The expression levels of NR4A1 were evaluated with the ΔΔCT method and normalized to the expression of the housekeeper GUSB (β-glucuronidase).

Primer sequences (5′ to 3′):

      forward            reverse

NR4A1  GTGTGTGGGGACAACGCTTC CCACAGGGCAGTCCTTGTT

GUSB     GACACCCACCACCTACATCG CTTAAGTTGGCCCTGGGTCC

### 2.11. Computational Analysis

The Cytoscape v.3.7.1 software was implemented to generate the human gene–miR interaction pathways linked to the searched miRs (miR-15, miR 26a, miR-21, and miR-146a) and proinflammatory cytokines (INFγ, IL-2, IL-4, IL-6, IL-10, TNFα, EGF, IL-17, IL-17A, MIP3A, CCL20/MIP3A, and RANTES). The miRTarBase and TargetScan databases were used for the miR–target gene interaction types together with a list of the searched miRs by using the CyTargetLinker app v4.10 [43]. For the gene–gene interaction network containing proinflammatory cytokines, the GeneMANIA algorithm was applied, comprising 911 networks and 24,280 genes [44,45]. Finally, the merge tool as a part of the Cytoscape pipeline was implemented to merge previously generated networks combining analyzed miRs and cytokines.

### 2.12. Analysis and Statistics

Data are shown as mean and standard deviation (SD) of biological independent experiments. One-way ANOVA with Dunnett’s multiple comparison test or Tukey’s *t*-Test, as indicated in the figure legends, was performed to evaluated the statistical significance between untreated und treated cells with GraphPad Prism 8. *p* < 0.05 was considered significant, * *p* < 0.05, ** *p* < 0.01, *** *p* < 0.001.

## 3. Results

### 3.1. Quantification of Catecholamines in Serum of Patients with Different Heart Diseases

The CAT noradrenaline, adrenaline, and dopamine were measured in the serum of postmenopausal healthy patients and patients admitted with TTS (acute and subacute phase). Significantly elevated noradrenaline (5.23 ± 5.69 pmol/L) vs. (4.24 ± 1.52 pmol/L) and dopamine (0.80 ± 1.06 pmol/L) vs. (0.18 ± 0.08 pmol/L) levels were detected in the TTS subacute group exclusively; all other levels did not differ significantly from the control group, postmenopausal healthy females (Table 1). Among the three CATs, noradrenaline showed highest absolute serum levels among the groups, while dopamine and adrenaline varied. 

In the following, endothelial cells and cardiomyocytes were treated in vitro with the different CAT concentrations as determined from TTS acute and TTS subacute patient serum as opposed to pharmacological concentrations (concentration c1 pharmacological concentrations [42]: 150 µM dopamine, 1 µM epinephrine, 1 µM norepinephrine; c2 postmenopausal healthy control; c3 TTS acute; c4 TTS subacute, comp. Table 1).

### 3.2. The Concentration-Dependent Effects of CAT on cEND and MyEND Brain and Myocardial Endothelial Cells In Vitro

To investigate the concentration-dependent change in the protein expression of endothelial-barrier-forming proteins claudin-5, occludin, VE-cadherin, and ZO-1 following CAT exposure in vitro, cerebral cEND cells were incubated with the different concentrations of the CAT dopamine, epinephrine, and norepinephrine as determined from patient serum comp. Table 1 compared to an exposure to supraphysiological/pharmacological concentrations [46] for 24 h (Figure 1).

Changes in protein levels were determined using Western blot analysis and comparing results to those of the untreated control group (Figure 1). For cEND cells, the supraphysiological/pharmacological c1 concentration showed significant effects on the protein expressions: ZO-1, VE-cadherin, and occluding expression were significantly reduced (0.451 ± 0.130-fold) and (0.186 ± 0.069-fold). Surprisingly, claudin-5 expression was significantly increased (1.761 ± 0.312-fold). cEND monolayer exposure to CATs in concentration ranges as determined from TTS patient serum (c2, c3, c4) showed no significant change in the protein expression (Figure 1).

The concentration-dependent changes in mRNA expression were determined using qRT-PCR analysis and comparing results to those of the untreated control group (Figure 2). Only the supraphysiological/pharmacological c1 concentration showed significant effects on the mRNA expression. Claudin-5 expression was significantly reduced (0.610 ± 0.059-fold). Additionally, VE-cadherin and ZO-1 showed a significant reduction (0.582 ± 0.139-fold) and (0.478 ± 0.085-fold). The exposure of cEND monolayers with CATs in concentrations as determined from TTS patient serum (c2, c3, c4) and occludin showed no significant change in the mRNA expression (Figure 2).

For MyEND cells, comparably, only the supraphysiological/pharmacological c1 concentration showed significant effects on the protein expressions: Claudin-5 expression levels were significantly decreased in this application (0.275 ± 0.067-fold), and occludin expression was significantly reduced (0.686 ± 0.122-fold). The other CAT concentrations applied as determined from TTS patient serum (c2, c3, c4) showed no significant change in the protein expression of claudin-5 or occluding.

The concentration-dependent changes in mRNA expression were equally determined by RT-qPCR analysis and compared to the untreated control group (data not shown). Claudin-5 (0.312 ± 0.09-fold) and occludin (0.473 ± 0.055-fold) expression were significantly reduced after incubation with CAT in concentration c1 (pharmacological). Occludin expression was significantly reduced as well after incubation with c1. VE-cadherin and ZO-1 showed a significant reduction after c1 incubation (0.438 ± 0.020-fold) and (0.463 ± 0.0259-fold), respectively.

### 3.3. The Concentration-Dependent Effects of CATs on iPSC-Derived Cardiomycytess

To investigate the effects of physiological concentrations of CATs (CAT) in TTS patients on the myocardium, we treated human iPSC-derived cardiomyocytes with CATs for 24 h. To determine changes in cardiomyocytes upon CAT treatment, we analyzed the expression level of nuclear receptor subfamily 4 group A member 1. This hormone receptor is well expressed in the heart and upregulated after β-adrenergic stimulation [12]. Additionally to the CATs, cardiomyocytes were treated with a supraphysiological concentration of isoprenaline (Iso). As reported by Borchert et al., the treatment with Iso lead to a significant increase in the cardiac stress marker, whereas physiological CAT concentrations showed no alterations in cardiomyocyte stress levels (Figure 3A). Multiples of the physiological CAT concentration (10^3^ and 10^6^ of the physiological concentration) led to cardiac stress indicated by increased expression of NR4A1 (Figure 3B); values 10^9^-fold that of the physiological concentration were lethal for the cardiomyocytes.

### 3.4. Quantification of Cytokines and Chemokines in Patient Serum

A previous study demonstrated that treatment with inflammatory cytokines (TNFα, Interleukin-6) leads to compromised endothelial barrier function and reduced or altered junctional protein expression established in vitro models of the BBB, cEND cells [47], and in vivo studies [48]. Based on these findings, we hypothesized that inflammation is central to the pathophysiology of TTS and explored the time course and persistence of detectable potentially involved proinflammatory cytokines. We hypothesized that inflammation is central to the pathophysiology and natural history of Takotsubo syndrome.

We compared the concentration of different cytokines and chemokines in the sera of premenopausal healthy, postmenopausal healthy, TTS acute, TTS subacute, TTS chronic, and ACS patients (Table 2). Although no statistical significance was found, the concentrations of TNFα, IL-2, IL-4, and IL-6 show a tendency to be elevated in the TTS acute group compared to all other groups. In comparison, in the TTS subacute group, the concentrations of MIP3A, RANTES, EGF, and IL-17 show a tendency to be elevated compared to all other groups. INFγ and IL-10 show higher levels in the postmenopausal healthy group compared to the other groups.

### 3.5. Quantification of Testosterone and Estradiol in Serum of Patients with Various Heart Diseases and Correlation between Testosterone-to-Estradiol Ratio

The sex hormones testosterone and estradiol were measured in the serum of healthy postmenopausal patients and patients with TTS (acute, subacute, and chronic). The different hormone levels were compared to healthy postmenopausal patients. There was no significant difference between the groups detectable (Table 3).

The following ratio between testosterone and estradiol (T/E2) as measured in the serum of patients with various heart diseases was calculated: mean T/E postmeno [5.77 + 0.2]; premeno diestrous [0.5 ± 0.1]; postmeno estrous [8.1 ± 3.3]; TTS acute [32 ± 0.4]; TTS subacute [6.7 ± 0.1]; and TTS chronic [16.11 ± 0.3]. As reference values for healthy premenopausal females, the E2 values estrous/di-estrous (400 pg/mL/50 pg/mL) were used, respectively (University of Rochester, medicine labs).

The calculated T/E2 ratio was the highest in acute TTS serum (32-fold), followed by chronic TTS (26-fold) and subacute TTS (8-fold). The lowest T/E2 ratios were measured in the healthy groups.

### 3.6. Protein Expression of Glucocorticoid Receptor α in Exosomes of Serum from Patients with Various Heart Diseases

Glucocorticoids are primary stress hormones necessary for life that regulate numerous physiologic processes in an effort to maintain homeostasis [49]. For the determination of GRαproteins levels, after the extraction of the exosomes of serum from patients with various heart diseases, Western blot analysis of glucocorticoid receptor α was performed. Changes in protein levels were compared to the healthy postmenopausal group, showing a clear trend towards elevation in all TTS groups, remarkably, the TTS acute and TT chronic groups. Remarkably, even though they were not significant, values determined from female TTS patients approximate much higher serum GR levels than males (1.59 of healthy female premenopausal), determined from the baseline values of healthy females (postmenopausal healthy set = 1; premenopausal healthy = 0.7 ± 0.55) (Figure 4).

### 3.7. Bioinformatic Analysis

Next, we set out to relate the identified inflammatory cytokines present in TTS patient blood to published miRNA in TTS, aging and age-related disease and inflammaging, and the CATs described to play a role in the etiology of TTS [5,6,7,8,22,23,50,51]. A potential miRNA–cytokine–steroid hormone-receptor regulatory network contributing to TTS was developed.

The Cytoscape algorithm was able to reconstitute a detailed GMI network comprising 2989 nodes and 6000 edges associated with the corresponding miRs by utilizing the miRTarBase and TargetScan databases (Figure 5) [44].

Next, the GGI network was built using the GeneMANIA algorithm for the corresponding blood markers and proinflammatory cytokines comprising 211 nodes and 626 edges (Figure 6) [45].

Finally, the merged network was compiled from the GMI and GGI networks containing 3152 nodes and 5735 edges to connect the blood markers, proinflammatory cytokines, and glucocorticoid receptor gene (NR3C1) to miRs (Figure 7A) [44]. This network was processed by subtracting the analyzed nodes to identify only direct gene–miR interactions (Figure 7B). As a result, IL-6, CCL5, CCL20, and INFγ were predicted to interact with the analyzed miRs directly, while the rest of the genes were involved in indirect interaction with the investigated miRs.

## 4. Discussion

In clinical practice, a few blood biomarkers are used to diagnose and monitor TTS relative to ACS, which is characterized by almost the same clinical presentations and ST elevations. These biomarkers serve to predict response to treatment as well as long-term prognosis; blood troponin T, BNP, or creatine kinase-MB (CK-MB) are the most frequently surveyed markers. [52,53] These markers together with clinicopathological findings [54,55] convey insufficient diagnostic accuracy.

### 4.1. TTS Patient Serum Catecholamine Levels

Takotsubo patients present with supraphysiologic plasma CAT levels (epinephrine, norepinephrine, dopamine), which is generally believed to be a cardiotoxic trigger [10,11] causal for the abnormal LV contraction pattern in TTS [10,11,56]. Available information, however, remains controversial [9,57] and heavily relies on disease modeling in a dish [34] or using the isoprenaline rodent model of TTS [42] creating a supraphysiological CAT exposure for the cell systems or animals investigated while CAT effects on the cardiovascular system in narrower physiological ranges remain elusive. Supporting the CAT hypothesis, some cross-field references can be found in the literature, e.g., pheochromocytoma-related conditions is a CAT-mediated myocarditis and focal/diffuse myocardial fibrosis condition [58]. Contrast to this hypothesis, our in vitro assessments did not point to structural endothelial cell (from BBB, myocard) or cardiomyocyte damage when exposed to CAT in concentrations determined from TTS patient blood.

Indirect CAT effects on the cardiovascular system in the pathogenesis of TTS should therefore be considered; meanwhile, it has been acknowledged that immune and central nervous systems mutually influence one another in a bidirectional way, e.g., in the physiological stress response [59]. Specifically, the contribution of the locus coeruleus, the principal site for the brain synthesis of norepinephrine in the genesis of defeat-stress-induced inflammatory signaling, should be mentioned [60].

To conclude, CAT serum levels a reliable TTS biomarker; further research is needed to examine the link between CATs and TTS disease ontogeny. In the following, we thus concentrate on the elucidation of disparate novel blood-borne biomarkers of stress and sex differences and inflammation in this stress-related disease.

### 4.2. TTS Patient GR Serum Levels

We did observe a consistent trend towards elevated GR levels in TTS patient serum. Measured GR serum levels in TTS patient blood clearly exceeded the average postmenopausal healthy female levels by 1.6-fold and came close to the values determined from male blood donors.

Generally, stress responses are inhibited by negative feedback mechanisms, whereby glucocorticoids act to diminish drive (brainstem) and promote transsynaptic inhibition via limbic structures (e.g., hippocampus), including the feedback downregulation of GR [61]. In contrast, elevated GR [62,63] expression has been described as a suitable surrogate marker and to positively correlate with glucocorticoid resistance in chronic stress [64]. Chronic stress-induced activation of the hypothalamus–pituitary–adrenal axis takes many forms (chronic basal hypersecretion, sensitized stress responses, and even adrenal exhaustion), with manifestation dependent upon factors such as stressor chronicity, intensity, frequency, and modality which involves GR insensitivity and upregulation [61]. Neural mechanisms driving acute and chronic stress responses can be distinct by the serum GR expression, taking into account age, sex, and reproductive cycle biases [65].

The activation of the HPA axis and sympathetic nervous system overdrive, which are established mechanisms critically involved in the TTS ontogeny, are widely accepted to link specifically chronic stress with elevated levels of peripheral proinflammatory markers in blood [66]. However, empirical evidence showing that peripheral levels of glucocorticoids and/or CATs mediate this effect is equivocal. Recent attention has turned to the possibility that cellular sensitivity to these ligands may enhance the effects of inflammatory mediators released under chronic stress.

The glucocorticoid resistance in conjunction with β2-adrenergic receptor signaling pathways in neuronal cells was presented to promote peripheral proinflammatory conditions that were postulated to be associated with chronic psychological stress [67].

However, given the limited size of our study and patient cohort, further research is warranted to examine the link between serum stress, GRα levels, and TTS disease progression in female postmenopausal patients.

### 4.3. Inflammation: Cytokines and Chemokines in Patient Serum

The existence of an immune–endocrine relation has been described, pointing out the immuno-modulatory effects of the CATs epinephrine, norepinephrine, and dopamine on a wide variety of immune functions [68]. The authors described that the immunomodulatory effects are mediated by specific adrenergic and dopaminergic receptors detectable on the surface of their immunological target cells and that CATs released within the CNS may chiefly influence the activity of the immune system in the periphery [68]. We explored the expression levels and time course of the expression of various pro- and anti-inflammatory cytokines while comparing their levels to healthy pre- and postmenopausal subjects plus ACS patients.

Patients with acute TTS had higher serum concentrations of IL-6, CXCL1 (growth-regulated protein) chemokine, TNFα, INFγ, IL-2, and IL-4 compared with control subjects and acute ACS patients, respectively, while EGF-levels were decreased. This observation was in agreement with prior observations published by Scally et al. [27] and Santoro, [69] which also compared inflammatory patterns in TTS and ACS.

Based on the data presented, monitoring the elevation of the cytokines IL-6, TNFα, and INFγserum levels might be suitable to distinguish TTS acute from ACS, again highlighting the importance of inflammation in the etiology of TTS. Novel acute phase markers identified include the elevation of both IL-2 and IL-4. The exploitability of IL-2 (increase) and IL-10 (decrease) as both disease phase markers and distinction markers of ACS and TTS needs further exploitation in bigger patient cohorts and warrants future potential as relapse indicators.

The evaluation of IL-2 could represent a novel, TTS acute phase marker, facilitating the discrimination from ACS. Specifically, as IL-2 has been characterized as an anti-inflammatory cytokine, the determination of its levels from patient serum might be a tool to predict propensity to relapse [70]. However, taking into account the limited number of samples available, a follow-up study including higher sample numbers would be mandatory before final conclusions could be drawn about the suitability of IL-2 as a TTS acute phase marker or to differentiate TTS from ACS, or its potential suitability as a relapse marker.

The subacute cytokine profile is most distinctively characterized by elevations in MIP3A, a so far not yet addressed cytokine in frame of TTS, in EGF, TNFα, IL-17, and, interestingly, pronounced decreases in IL-10. Amongst those, IL-17 might represent a future therapeutic target in TTS: Interleukin-17 (IL-17) induces the production of granulocyte colony-stimulating factor (G-CSF) and chemokines such as CXCL1 and CXCL2, and is a cytokine that acts as an inflammation mediator [71].

Taken together, while the elevation of IL-6, IL-2, IL-4, TNFα, INFγ, and CXCL1 can be regarded as TTS acute phase serum markers, a sharp increase in MIP3A and a pronounced decrease in IL-10 was detectable in the TTS subacute phase, which culminated in the chronic phase but also in ACS. In contrast, IL-10 levels were the highest in the postmenopausal healthy group.

### 4.4. Sex Differences

The epidemiological link between female sex and the incidence of TTS cannot be fully explained by stress-induced CAT or corticosteroid levels only. Additional factors, such as female sex or hormone receptor regulation and dysregulated immune response, could be an important factor for this specific form of heart failure in postmenopausal patients. Consistent with prior studies, we found strong associations between reduced female sex steroid levels and further extended these studies by calculating a T/E2 ratio which was elevated pronouncedly in TTS patients. Several points could be taken into account:

Women mount stronger immune responses not only against foreign but also against self-antigens, and the prevalence of most autoimmune diseases (AD) is greatly increased in women compared to men. An important role underlying the difference in activity of immune cells in men and women is attributed to sex hormones [4,5].

Sexually dimorphic actions of glucocorticoids have been shown to provide a link to inflammatory diseases with sex differences in prevalence [59]. Our data strongly resonate with the findings of Walsh et al. [67], who reported the consistent observation of elevated levels of peripheral proinflammatory markers accompanying chronic stress in humans, which associates with not only heightened GR expression, an assumed result of decreased cellular GR sensitivity, and alterations in ß-AR signaling, which was interpreted to be based on myeloid-progenitor-cell-related stress signaling pathways [67]. This situation could also be associated with TTS pathophysiology. Lastly, the exact role that the T/E2 ratio plays in the development of TTS remains unclear and warrants further investigation.

### 4.5. Bioinformatic Extrapolation: TTS—An Inflammaging Disorder?

In the cardiovascular system, and like many other vital organs, the heart is susceptible to the effects of aging. Inflammation is a hallmark of aging and other coexisting comorbidities linked to age-related decline, such as heart failure, cardiovascular disease, neurodegeneration, vascular dementia, and Alzheimer’s disease, but also age-related hearing loss, age-related macular degeneration, and type 2 diabetes [28,72,73,74]. Inflammation is also associated with TTS. Recently, a novel portemanteau of inflammation and aging has been coined, “inflammaging”, which is defined as chronic, low-grade inflammation accumulated and worsening over age, presumably contributing to the pathogenesis of various age-related pathologies. [73,75,76] Inflammation has been described to stem from the cellular senescence of the immune system, so-called immunosenescence [77]. Considering inflammatory failure [78], we set out to lay attention on the role of inflammaging in TTS. In humans and experimental animal models, several reports [27,69] have documented a significant association between four systemic markers of inflammation, i.e., IL-6, IL-10, TNFα, and INFγ [27,34,42,69].

Our computational analyses predicted IL-6, CCL5, CCL20, and INFγ to interact with the studied miRs directly, while the remaining genes were shown to be involved in indirectly with these miRs. In particular, possible INFγ signaling pathways were found to be relevant in the ontogeny of TTS (miR-15a and miR-15b). On the other hand, CCL5, CCL20, and IL-6 were predicted to play a pivotal role in inflammation (miR-21 and miR-146a). Similarly, the IL-6 contribution to the TTS lipidome was previously determined by experimental and computational analyses connecting this cytokine to the phosphatidylinositol molecule [79]. Interestingly, several studies have reported the increased expression of miR-21 and miR-142 in prostate cancer, highlighting their role as independent diagnostic factors and their association with enhanced cancer cell growth [80]. Furthermore, miR-146a inhibits vimentin expression, suppressing cell proliferation, migration, and invasion in esophageal squamous cell carcinoma, potentially linked to advanced stages and poor overall survival [81]. Additionally, miRNAs from the miR-15/16 cluster, including miR-15a and miR-16-1, act as tumor suppressors by inhibiting cell proliferation and promoting apoptosis, particularly in chronic lymphocytic lymphoma [82].

While age and sex are determinants of TTS [66,83,84], the neutrophile/lymphocyte ratio [85], renin–angiotensin system [86], and atrial fibrillation [87] are shown to be associated with prognosis of TTS. Amongst these well-described factors of TTS, inflammaging-related pathways add to the relevant risk factors or prediagnostic markers of TTS. Currently, the role of inflammaging in the ontogeny of TTS is only speculative, and further large-scale studies are needed to verify this.

To conclude, the contributions of various molecular mechanisms, notably, inflammaging, oxidative stress, CUMS, sex, and genetic factors, shall be evaluated as possible common culprits that interrelate the pathophysiology in the heart and cardiovascular system as part of TTS. Future research using high-throughput or omics techniques for the analysis of blood samples from larger real-life patient cohorts will probably gain better insight into the underlying mechanisms of different disease phenotypes. The identification of specific molecular pathways and associated biomarkers could then allow the development of new targets for precision medicine.

## Figures and Tables

**Figure 1 biomolecules-13-00995-f001:**
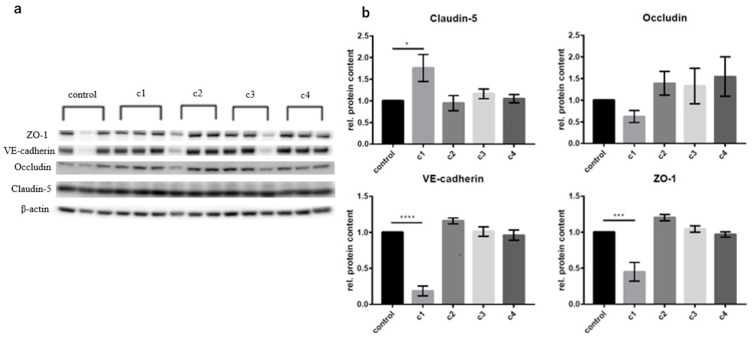
Changes in protein expression in cEND cells after incubation with the CATs norepinephrine, epinephrine, and dopamine in different concentration (c1 pharmacological concentrations [18]: 150 µM dopamine, 1 µM epinephrine, 1 µM norepinephrine; c2 postmenopausal healthy control; c3 TTS acute; c4 TTS subacute, comp. Table 1). (**a**) Western blots displaying expression of the different proteins after 24 h of incubation. (**b**) Densitometric evaluation. Changes in protein expression was normalized to β-actin. Data are the means (±SEM) of 3 independent experiments. Statistical significance was evaluated using Dunnett’s multiple comparisons test. *: *p* < 0.05; ***: *p* < 0.001; ****: *p* < 0.00001.

**Figure 2 biomolecules-13-00995-f002:**
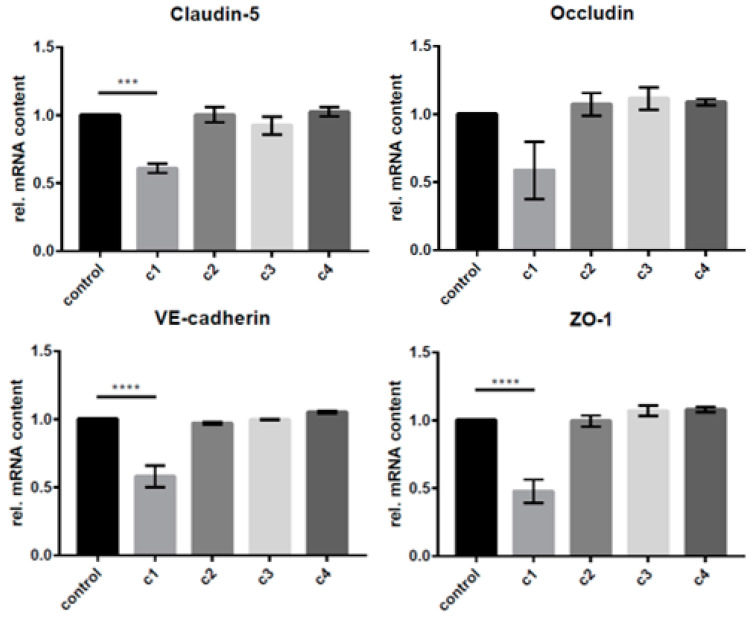
Changes in mRNA expression in cEND cells after incubation with the CATs norepinephrine, epinephrine, and dopamine in different concentrations (c1 pharmacological concentrations [18]: 150 µM dopamine, 1 µM epinephrine, 1 µM norepinephrine; c2 postmenopausal healthy control; c3 TTS acute; c4 TTS subacute, comp. Table 1). Changes in protein expression were normalized to β-actin. Data are the means (±SEM) of 3 independent experiments. Statistical significance was evaluated using Dunnett’s multiple comparisons test. ***: *p* < 0.001; ****: *p* < 0.00001.

**Figure 3 biomolecules-13-00995-f003:**
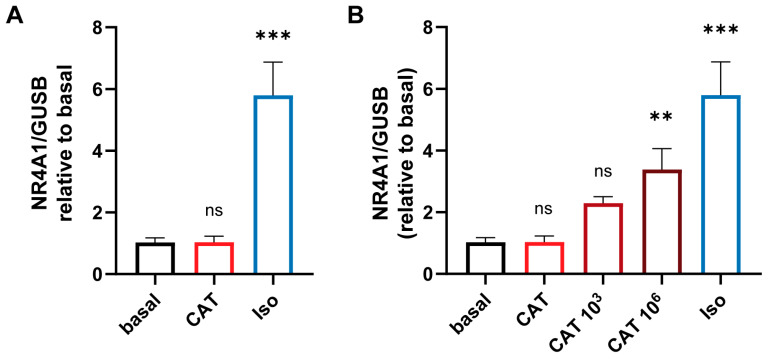
Physiological CAT concentrations did not alter cardiac stress marker NR4A1 in cardiomyocytes. (**A**) Human iPSC-derived cardiomyocytes were treated with physiological concentrations of a CAT mix as determined from acute TTS (c3 comp. Table 1) and supraphysiological concentrations of isoprenaline (Iso) for 24 h. The expression level of the cardiac stress marker nuclear receptor subfamily 4 group A member 1 (NR4A1) was analyzed by RT-qPCR; mean ± SD of three independent experiments is shown. (**B**) Human iPSC-derived cardiomyocytes were treated with physiological concentrations of CAT as determined from acute TTS (c3) and its multiples (10^3^, 10^6^). Expression of NR4A1 was determined by RT-qPCR; mean ± SD of triplicates is shown. Statistical significance was analyzed with one-way ANOVA with Dunnett’s correction in comparison to basal, ** *p* < 0.01, *** *p* < 0.001, ns *p* > 0.05.

**Figure 4 biomolecules-13-00995-f004:**
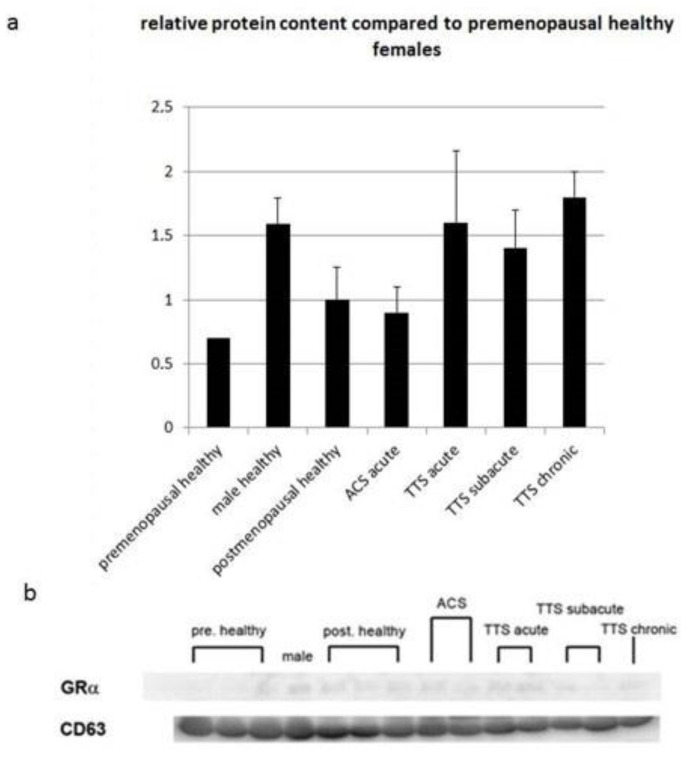
Changes in the protein expression of glucocorticoid receptor α in exosomes from serum from patients with various heart diseases. (**a**) Expression of glucocorticoid receptor α in the exosomes (densitometric evaluation). (**b**) Membranes of the Western blots used in this experiment. Glucocorticoid receptor α protein expression was normalized to CD63. Data are the means (±SD) of *n* independent patients (postmenopausal healthy: *n* = 3, premenopausal healthy: *n* = 3, ACS acute: *n* = 4, TTS acute: *n* = 4, TTS subacute: *n* = 8, TTS chronic: *n* = 3). Statistical significance was evaluated using Tukey’s multiple comparisons test.

**Figure 5 biomolecules-13-00995-f005:**
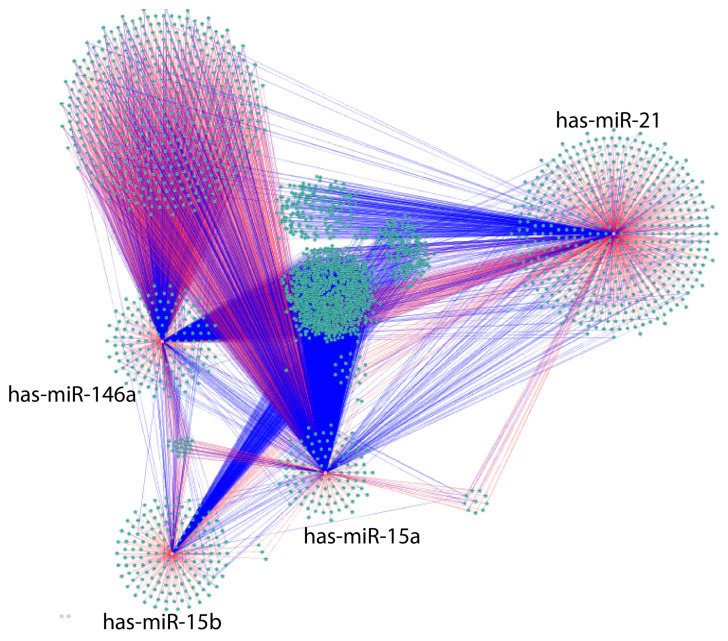
Human GMI (gene–miR interaction) network generated by the CyTargetLinker algorithm human using the miRTarBase (blue) and TargetScan (red) databases. The marked nodes are depicted in yellow. The network is displayed using the yFiles organic layout.

**Figure 6 biomolecules-13-00995-f006:**
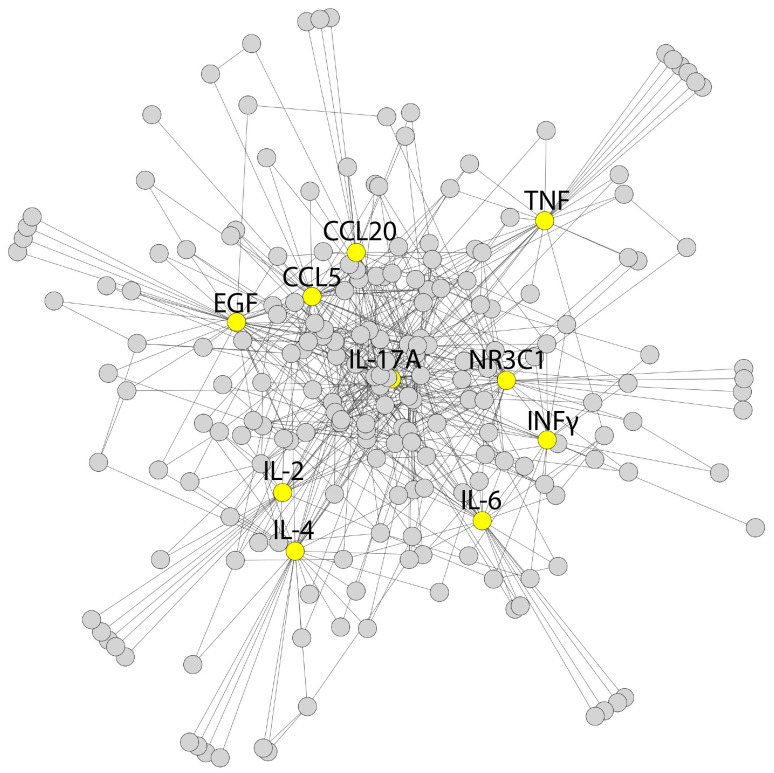
Human GGI (gene–gene interaction) network generated by the GeneMANIA plugin for the corresponding blood markers and proinflammatory cytokines (CCL20, CCL5, EGF, IL2, IL4, IL6, IL10, IL17A, INFγ, NR3C1, and TNF). The marked nodes are depicted in yellow. The network is displayed using the yFiles organic layout.

**Figure 7 biomolecules-13-00995-f007:**
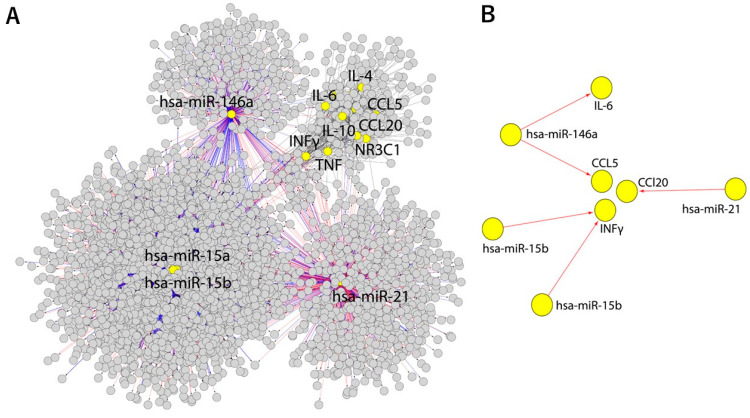
The merged (**A**) and subtracted (**B**) networks produced by merging GMI and GGI networks or creating a new network (8 nodes and 5 edges) from selected nodes and all of their edges. The marked nodes are depicted in yellow. The merged network is displayed using the perfuse force-directed layout.

**Table 1 biomolecules-13-00995-t001:** Measured concentration of different CATs (CAT) in serum of postmenopausal healthy patients and patients with TTS in pmol/L ± SD. *: *p* < 0.0); [TTS acute] = c2, [TTS subacute] = c3 in the following sections.

	Postmenopausal Healthy *n* = 3	TTS Acute *n* = 4	TTS Subacute *n* = 8
Noradrenaline	4.24 ± 1.52	3.65 ± 2.37	5.23 ± 5.69 *
Adrenaline	0.22 ± 0.11	0.13 ± 0.1	0.19 ± 0.2
Dopamine	0.18 ± 0.08	0.21 ± 0.12	0.80 ± 1.06 *

**Table 2 biomolecules-13-00995-t002:** Measured concentration of different cytokines and chemokines in patient sera in pg/mL ± SD.

	Premenopausal Healthy *n* = 3	Postmenopausal Healthy *n* = 3	TTS Acute *n* = 4	TTS Subacute *n* = 8	TTS Chronic *n* = 3	ACS *n* = 4
INFγ	2.41 ± 1.48	18.7 ± 7.18	65.6 ± 71.99	26.79 ± 32.46	19.81 ± 30.50	18.46 ± 32.03
IL-2	0.32 ± 0.33	1.49 ± 1.23	4.87 ± 8.012	2.2 ± 3.263	1.53 ± 2.243	0.37 ± 0.3641
IL-4	2.34 ± 3.38	1.59 ± 0.88	50.77 ± 8.68	2.75 ± 4.29	0.80 ± 1.10	0.70 ± 0.92
IL-6	0.34 ± 0.34	6.64 ± 15.41	38.01 ± 45.06	8.98 ± 15.13	1.45 ± 1.45	0.97 ± 0.77
IL-10	12.13 ± 19.54	30.21 ± 27.33	24.48 ± 17.52	10.55 ± 10.74	3.88 ± 6.68	3.41 ± 5.79
TNFα	15 ± 5.67	18.53 ± 18.51	41.04 ± 47.72	25.8 ± 16.62	17.02 ± 5.262	15.6 ± 12.43
		Postmenopausal healthy *n* = 3	TTS acute *n* = 4	TTS subacute *n* = 8	TTS chronic *n* = 3	ACS *n* = 4
EGF		57.05	22.19	85.14	26.14	47.85
IL-17		3.40 ± 1.70	1.91 ± 0.84	25.67 ± 52.31	3.13	3.52
MIP3A		147.3 ± 104.4	111.3 ± 68.08	249.6 ± 168.5	105.3	72.69
		Postmenopausal healthy *n* = 3	TTS acute *n* = 4	TTS subacute *n* = 8		
RANTES		161,542 ± 190,762	69,217 ± 81,541	264,518 ± 381,721		

**Table 3 biomolecules-13-00995-t003:** Measured concentrations of testosterone and 17β-estradiol in pg/mL in the serum of patients with various heart diseases. Data are the means (±SD) of *n* independent measures. Statistical significance was evaluated using Tukey’s multiple comparisons test.

	Postmenopausal Healthy *n* = 3	TTS Acute *n* = 4	TTS Subacute *n* = 8	TTS Chronic *n* = 3
Testosterone	202.5 ± 120.4	160	287.5 ± 257.5	145 ± 63.6
Estradiol	35.25 ± 54.04	5	43.13 ± 66.69	9.5 ± 6.364

## Data Availability

The original contributions presented in this study are included in the article, further inquiries can be directed to the corresponding authors.

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
