# Peer review of "Blood Biomarkers in Takotsubo Syndrome Point to an Emerging Role for Inflammaging in Endothelial Pathophysiology"

_biomolecules, 2023, doi:10.3390/biom13060995_

Round 1

Reviewer 1 Report

The manuscript by Nagai M et al. presents data suggesting that blood biomarkers in patients with Takotsubo syndrome may provide a very preliminary result for risk stratification and evaluation of disease. The study enrolled a small number of cohorts, and try to validate the results in animal models. The results are interesting and informative. The paper is overall well-written. The novelty of the study is average.

I think this is a topic that will allow us to unveil which is the contribution of the TTS assessment. However, is important to be aware and discussed that miR-15, miR-21, and miR-146a are established tumor/malignancy markers.

The introduction needs more work as well to describe the current understanding related to TTS, there are no statements of demographics in the current manuscript, and there is an absence of explanation on how the inflammation or what molecular pathways are involved?

The definition of serval concepts and the process of the experiment needed to be further clarified for rigor and repeatability.

When describing such a small number of patients, the percentage should be avoided (only 25 subjects).

The assumption and description of the results should be very careful, some of the statements in the discussion are not well-supported by the current results and references, especially in a relatively small number and poor level of evidence.

For the analysis of TTS, the result did provide certain information in this aspect, however, the mixed results and the meaning were lost in discussion. properly integrating this information are essential for presenting such a point of view.

There were a number of language and typos in the manuscript. Please review and correct the main text. (Line 114: und; etc.)

Author Response

Ms. biomolecules-2419031: "Blood biomarkers in Takotsubo syndrome point to an emerging role for inflammaging in endothelial pathophysiology"

Response to the Reviewer 1

We are very grateful for your comments, which have substantially improved our manuscript. We considered the following points and summarized our responses.

Major comments

  1. Comment: The manuscript by Nagai M et al. presents data suggesting that blood biomarkers in patients with Takotsubo syndrome may provide a very preliminary result for risk stratification and evaluation of disease. The study enrolled a small number of cohorts, and try to validate the results in animal models. The results are interesting and informative. The paper is overall well-written. The novelty of the study is average. I think this is a topic that will allow us to unveil which is the contribution of the TTS assessment.

Response: Thank you. We believe that this study provides some new insight of clinical relevance.

  1. Comment: However, is important to be aware and discussed that miR-15, miR-21, and miR-146a are established tumor/malignancy markers.

Response: Several studies have reported increased expression of miR-21 and miR-142 in prostate cancer, highlighting their role as independent diagnostic factors and their association with enhanced cancer cell growth (Bolayirli et al., 2022). Furthermore, miR-146a inhibits vimentin expression, suppressing cell proliferation, migration, and invasion in esophageal squamous cell carcinoma, potentially linked to advanced stages and poor overall survival (Chang et al., 2020). Additionally, miRNAs from the miR-15/16 cluster, including miR-15a and miR-16-1, act as tumor suppressors by inhibiting cell proliferation and promoting apoptosis, particularly in chronic lymphocytic lymphoma (Aqeilan et al., 2010).

Bolayirli, I. M., et al. (2022). "The clinical significance of circulating miR-21, miR-142, miR-143, and miR-146a in patients with prostate cancer." J Med Biochem 41(2): 191-198.

Chang, H. Y., et al. (2020). "MicroRNA-146a suppresses tumor malignancy via targeting vimentin in esophageal squamous cell carcinoma cells with lower fibronectin membrane assembly." J Biomed Sci 27(1): 102.

Aqeilan, R. I., et al. (2010). "miR-15a and miR-16-1 in cancer: discovery, function and future perspectives." Cell Death Differ 17(2): 215-220.

  1. Comment: The introduction needs more work as well to describe the current understanding related to TTS, there are no statements of demographics in the current manuscript, and there is an absence of explanation on how the inflammation or what molecular pathways are involved TTS?

Response: A study comparing TTS with myocardial infarction (MI) found that the former pathology primarily affects women and has distinct demographic and co-morbid predictors, including a higher prevalence of cerebrovascular accidents, drug abuse, anxiety disorders, mood disorders, malignancy, chronic liver disease, and sepsis. These findings highlight the importance of considering these factors when diagnosing SC and differentiating it from MI (El-Sayed et al., 2012). 

El-Sayed, A. M., et al. (2012). "Demographic and co-morbid predictors of stress (takotsubo) cardiomyopathy." Am J Cardiol 110(9): 1368-1372.

In a multi-center study, patients with TTS showed increased inflammation markers, including ultrasmall superparamagnetic particles of iron oxide (USPIO) enhancement in the myocardium, elevated serum interleukin-6 concentrations, and altered blood monocyte subpopulations compared to control subjects. However, these inflammatory changes resolved at 5 months follow-up, except for persistent elevations in serum interleukin-6 concentrations and reductions in intermediate monocyte subpopulations (Scally et al., 2019).

Scally, C., et al. (2019). "Myocardial and Systemic Inflammation in Acute Stress-Induced (Takotsubo) Cardiomyopathy." Circulation 139(13): 1581-1592.

In a study by Nef et al., they examined endomyocardial biopsy samples from 16 TTS patients and observed increased activation of PI3K and Akt pathways, which were later found to return to normal levels during follow-up and were not present in cases of ischemia (Nef et al., 2009). These pathways are associated with β2AR-Gαi activity and play a role in promoting the survival of cardiomyocytes. Furthermore, Nakano et al. found elevated levels of proteins involved in β2AR-Gαi signaling, including GRK and β-arrestin, in tissue samples from 26 patients with acute TTS, highlighting their importance in facilitating β2AR-Gαi stimulus trafficking (Nakano et al., 2018).

Nef, H. M., et al. (2009). "Activated cell survival cascade protects cardiomyocytes from cell death in Tako-Tsubo cardiomyopathy." Eur J Heart Fail 11(8): 758-764.

Nakano, T., et al. (2018). "Alteration of beta-Adrenoceptor Signaling in Left Ventricle of Acute Phase Takotsubo Syndrome: a Human Study." Sci Rep 8(1): 12731.

  1. Comment: The definition of serval concepts and the process of the experiment needed to be further clarified for rigor and repeatability.

Response: Yes, thank you for indicating the important issue. All of the comments substantially improve our study methods and results. As the reviewer indicated, we respond to Comment 4-6, as follows.

  1. Comment: When describing such a small number of patients, the percentage should be avoided (only 25 subjects).

Response: Yes, we changed: “Female gender was at a percentage (78% vs 100%, p=0.23) between the group with TTS and control.” to “¾ of the TTS patients were of female gender.”Pg3.

  1. Comment: The assumption and description of the results should be very careful, some of the statements in the discussion are not well-supported by the current results and references, especially in a relatively small number and poor level of evidence.

Response: We tried to point out that further research is needed:

Pg. 15 To conclude, CAT serum levels a reliable TTS biomarker, further research is needed to examine the link between CATs and TTS disease ontogeny.

Pg. 15 However, given the limited size of our study and patient cohort, further research is warranted to examine the link between serum stress, GR levels and TTS disease progression in female postmenopausal patients.

Pg. 16 added: However, given the limited size of our study and patient cohort, further research is warranted to establish the investigated cytokines as markers for the diverse disease phases.

Pg. 17: Specifically in light of the small patient cohort number examined, Pg. 17 addded: Currently, the role of inflammaging in the ontogeny of TTS is only speculative, and further large-scale studies are needed to verify. Was stated and highlighted again for better legiblity.

Conclusion pg 17 modified: To conclude, the contributions of various molecular mechanisms, notably inflammaging, oxidative stress, CUMS, gender, and genetic factors shall be evaluated as possible common culprits that interrelate the pathophysiology in the heart and cardiovascular system as part of TTS. Future research using high throughput

or omics techniques for the analysis of blood samples from bigger real-life patient cohorts will probably gain better insight into underlying mechanisms of different disease phenotypes. Identification of specific molecular pathways and associated biomarkers could then allow the development of new targets for precision medicine.

  1. Comment: For the analysis of TTS, the result did provide certain information in this aspect, however, the mixed results and the meaning were lost in discussion. properly integrating this information are essential for presenting such a point of view.

Response: We had modified the conclusion pg 17: To conclude, the contributions of various molecular mechanisms, notably inflammaging, oxidative stress, CUMS, gender, and genetic factors shall be evaluated as possible common culprits that interrelate the pathophysiology in the heart and cardiovascular system as part of TTS. Future research using high throughput or omics techniques for the analysis of blood samples from bigger real-life patient cohorts will probably gain better insight into underlying mechanisms of different disease phenotypes. Identification of specific molecular pathways and associated biomarkers could then allow the development of new targets for precision medicine.

As this is the first and – as pointed out reeatedly – small cohort, locally restricted (japan) study, no more exted conclusions can be drawn at this point. We believe this study is highly suitable to motivate future extended and in depth-follow up studies to foster a new doctrine.

  1. 7. Comment: Comments on the Quality of English Language. There were a number of language and typos in the manuscript. Please review and correct the main text. (Line 114: und; etc.)

Response: Thank you. We changed as follows;

Line114: “und” to “and”.

Line295: “parmacological” to “pharmacological”.

Line299: “as as” to “as”.

Line114: “und” to “and”.

Line295: “parmacological” to “pharmacological”.

Line306: “xpression” to “expression”.

Line314: “parmacological” to “pharmacological”.

Line329: “parmacological c1 concen tration” to “pharmacological c1 concentration”.

Line346: “cardiomycytes” to “cardiomyocytes”.

Line370: “Interleu kin-6” to “Interleukin-6”.

Line385: “patients” to “patient”.

Line412: “postmeno pausal” to “postmeno pausal”.

Line473: “characterised” to “characterized”.

Line593: “assumpted” to “assumed”.

Reviewer 2 Report

This study shows that inflammation-related pathways are identified to add to the relevant risk factors or pre-diagnostic markers of Takotsubo syndrome, hence is interesting and clinically significant for the patients with Takotsubo syndrome.

Importantly, in addition to age and gender (Murakami T, Komiyama T, Kobayashi H, Ikari Y. Gender Differences in Takotsubo Syndrome. Biology. 2022;11:653.; Pattisapu VK, Hao H, Liu Y, Nguyen TT, Hoang A, Bairey Merz CN, Cheng S. Sex- and Age-Based Temporal Trends in Takotsubo Syndrome Incidence in the United States. J Am Heart Assoc. 2021;10:e019583.; Nagai M, Förster CY, Dote K. Sex Hormone-Specific Neuroanatomy of Takotsubo Syndrome: Is the Insular Cortex a Moderator? Biomolecules. 2022;12:110.), other interesting factors as indicated by the following references should also be introduced or discussed.

[1]     Khan H, Rudd A, Gamble DT, Mezincescu AM, Cheyne L, Horgan G, Dhaun N, Newby DE, Dawson DK. Renin-Angiotensin and Endothelin Systems in Patients Post-Takotsubo Cardiomyopathy. J Am Heart Assoc. 2022;11:e025989.

[2]     El-Battrawy I, Cammann VL, Kato K, Szawan KA, Di Vece D, Rossi A, Wischnewsky M, Hermes-Laufer J, Gili S, Citro R, et al. Impact of Atrial Fibrillation on Outcome in Takotsubo Syndrome: Data From the International Takotsubo Registry. J Am Heart Assoc. 2021;10:e014059.

[3]     Zweiker D, Pogran E, Gargiulo L, Abd El-Razek A, Lechner I, Vosko I, Rechberger S, Bugger H, Christ G, Bonderman D, et al. Neutrophile-Lymphocyte Ratio and Outcome in Takotsubo Syndrome. Biology. 2022;11:1154.

Besides, there are many grammatical/spelling errors to be corrected. For example, in the Abstract:

“……glucocorticoid receptor (GR) expression levels were observed in the TTS patient collective.” should be “……glucocorticoid receptor (GR) expression levels were observed in the TTS patient collected.” Or “……glucocorticoid receptor (GR) expression levels were observed in the collective TTS patient.”.

“……such catecholamine concentrations did not damage the cellular integrity or function of neither endothelial cells forming the blood brain barrier, endothelial cells derived from myocardium nor cardiomyocytes in vitro.” should be “……such catecholamine concentrations did not damage the cellular integrity or function of either endothelial cells forming the blood brain barrier, endothelial cells derived from myocardium or cardiomyocytes in vitro.”.

“……factors of TTS (older age, female gender), inflammaging-related pathways were identified to……” should be “……factors of TTS (older age, female gender), inflammation-related pathways were identified to……”.

This study shows that inflammation-related pathways are identified to add to the relevant risk factors or pre-diagnostic markers of Takotsubo syndrome, hence is interesting and clinically significant for the patients with Takotsubo syndrome.

Importantly, in addition to age and gender (Murakami T, Komiyama T, Kobayashi H, Ikari Y. Gender Differences in Takotsubo Syndrome. Biology. 2022;11:653.; Pattisapu VK, Hao H, Liu Y, Nguyen TT, Hoang A, Bairey Merz CN, Cheng S. Sex- and Age-Based Temporal Trends in Takotsubo Syndrome Incidence in the United States. J Am Heart Assoc. 2021;10:e019583.; Nagai M, Förster CY, Dote K. Sex Hormone-Specific Neuroanatomy of Takotsubo Syndrome: Is the Insular Cortex a Moderator? Biomolecules. 2022;12:110.), other interesting factors as indicated by the following references should also be introduced or discussed.

[1]     Khan H, Rudd A, Gamble DT, Mezincescu AM, Cheyne L, Horgan G, Dhaun N, Newby DE, Dawson DK. Renin-Angiotensin and Endothelin Systems in Patients Post-Takotsubo Cardiomyopathy. J Am Heart Assoc. 2022;11:e025989.

[2]     El-Battrawy I, Cammann VL, Kato K, Szawan KA, Di Vece D, Rossi A, Wischnewsky M, Hermes-Laufer J, Gili S, Citro R, et al. Impact of Atrial Fibrillation on Outcome in Takotsubo Syndrome: Data From the International Takotsubo Registry. J Am Heart Assoc. 2021;10:e014059.

[3]     Zweiker D, Pogran E, Gargiulo L, Abd El-Razek A, Lechner I, Vosko I, Rechberger S, Bugger H, Christ G, Bonderman D, et al. Neutrophile-Lymphocyte Ratio and Outcome in Takotsubo Syndrome. Biology. 2022;11:1154.

Besides, there are many grammatical/spelling errors to be corrected. For example, in the Abstract:

“……glucocorticoid receptor (GR) expression levels were observed in the TTS patient collective.” should be “……glucocorticoid receptor (GR) expression levels were observed in the TTS patient collected.” Or “……glucocorticoid receptor (GR) expression levels were observed in the collective TTS patient.”.

“……such catecholamine concentrations did not damage the cellular integrity or function of neither endothelial cells forming the blood brain barrier, endothelial cells derived from myocardium nor cardiomyocytes in vitro.” should be “……such catecholamine concentrations did not damage the cellular integrity or function of either endothelial cells forming the blood brain barrier, endothelial cells derived from myocardium or cardiomyocytes in vitro.”.

“……factors of TTS (older age, female gender), inflammaging-related pathways were identified to……” should be “……factors of TTS (older age, female gender), inflammation-related pathways were identified to……”.

Author Response

Ms. biomolecules-2419031: "Blood biomarkers in Takotsubo syndrome point to an emerging role for inflammaging in endothelial pathophysiology"

Response to the Reviewer 2

We are very grateful for your comments, which have substantially improved our manuscript. We considered the following points and summarized our responses.

Major comments

  1. Comment: This study shows that inflammation-related pathways are identified to add to the relevant risk factors or pre-diagnostic markers of Takotsubo syndrome, hence is interesting and clinically significant for the patients with Takotsubo syndrome.

Response: Thank you. We believe that this study provides some new insight of clinical relevance.

  1. Comment: Importantly, in addition to age and gender (Murakami T, Komiyama T, Kobayashi H, Ikari Y. Gender Differences in Takotsubo Syndrome. Biology. 2022;11:653.; Pattisapu VK, Hao H, Liu Y, Nguyen TT, Hoang A, Bairey Merz CN, Cheng S. Sex- and Age-Based Temporal Trends in Takotsubo Syndrome Incidence in the United States. J Am Heart Assoc. 2021;10:e019583.; Nagai M, Förster CY, Dote K. Sex Hormone-Specific Neuroanatomy of Takotsubo Syndrome: Is the Insular Cortex a Moderator? Biomolecules. 2022;12:110), other interesting factors as indicated by the following references should also be introduced or discussed. [1] Khan H, Rudd A, Gamble DT, Mezincescu AM, Cheyne L, Horgan G, Dhaun N, Newby DE, Dawson DK. Renin-Angiotensin and Endothelin Systems in Patients Post Takotsubo Cardiomyopathy. J Am Heart Assoc. 2022;11:e025989. [2] El-Battrawy I, Cammann VL, Kato K, Szawan KA, Di Vece D, Rossi A, Wischnewsky M, Hermes-Laufer J, Gili S, Citro R, et al. Impact of Atrial Fibrillation on Outcome in Takotsubo Syndrome: Data From the International Takotsubo Registry. J Am Heart Assoc. 2021;10:e014059. [3] Zweiker D, Pogran E, Gargiulo L, Abd El-Razek A, Lechner I, Vosko I, Rechberger S, Bugger H, Christ G, Bonderman D, et al. Neutrophile-Lymphocyte Ratio and Outcome in Takotsubo Syndrome. Biology. 2022;11:1154.

Response: Thank you. This is very important issue. As the reviewer indicated, while age and gender are determinants of TTS (Murakami T, et al., 2022; Pattisapu, et al., 2021), neutrophile-lymphocyte ratio (Zweiker et al., 2022), renin-angiotensin system (Khan et al., 2022) and atrial fibrillation (El-Battrawy et al., 2021) are shown to be associated with prognosis of TTS.

Murakami T, Komiyama T, Kobayashi H, Ikari Y. Gender Differences in Takotsubo Syndrome. Biology. 2022;11:653.; Pattisapu VK, Hao H, Liu Y, Nguyen TT, Hoang A, Bairey Merz CN, Cheng S. Sex- and Age-Based Temporal Trends in Takotsubo Syndrome Incidence in the United States. J Am Heart Assoc. 2021;10:e019583

Pattisapu VK, Hao H, Liu Y, Nguyen TT, Hoang A, Bairey Merz CN, Cheng S. Sex- and Age-Based Temporal Trends in Takotsubo Syndrome Incidence in the United States. J Am Heart Assoc. 2021;10:e019583.

Zweiker D, Pogran E, Gargiulo L, Abd El-Razek A, Lechner I, Vosko I, Rechberger S, Bugger H, Christ G, Bonderman D, et al. Neutrophile-Lymphocyte Ratio and Outcome in Takotsubo Syndrome. Biology. 2022;11:1154.

Khan H, Rudd A, Gamble DT, Mezincescu AM, Cheyne L, Horgan G, Dhaun N, Newby DE, Dawson DK. Renin-Angiotensin and Endothelin Systems in Patients Post Takotsubo Cardiomyopathy. J Am Heart Assoc. 2022;11:e025989.

El-Battrawy I, Cammann VL, Kato K, Szawan KA, Di Vece D, Rossi A, Wischnewsky M, Hermes-Laufer J, Gili S, Citro R, et al. Impact of Atrial Fibrillation on Outcome in Takotsubo Syndrome: Data From the International Takotsubo Registry. J Am Heart Assoc. 2021;10:e014059.

In the line 622, we added: “While age and gender are determinants of TTS (Nagai, et al., 2022; Murakami, et al., 2022; Pattisapu, et al., 2021), neutrophile-lymphocyte ratio (Zweiker et al., 2022), renin-angiotensin system (Khan et al., 2022) and atrial fibrillation (El-Battrawy et al., 2021) are shown to be associated with prognosis of TTS. Amongst the well-described these factors of TTS, inflammaging-related pathways add to relevant risk factors or prediagnostic markers of TTS.”

Minor comments

  1. Comment: “……glucocorticoid receptor (GR) expression levels were observed in the TTS patient collective.” should be “……glucocorticoid receptor (GR) expression levels were observed in the TTS patient collected.” Or “……glucocorticoid receptor (GR) expression levels were observed in the collective TTS patient.”

Response: We changed to “in the TTS patient collected.”

  1. Comment: “……such catecholamine concentrations did not damage the cellular integrity or function of neither endothelial cells forming the blood brain barrier, endothelial cells derived from myocardium nor cardiomyocytes in vitro.” should be “……such catecholamine concentrations did not damage the cellular integrity or function of either endothelial cells 4 forming the blood brain barrier, endothelial cells derived from myocardium or cardiomyocytes in vitro.”.

Response: We changed to “……such catecholamine concentrations did not damage the cellular integrity or function of either endothelial cells 4 forming the blood brain barrier, endothelial cells derived from myocardium or cardiomyocytes in vitro.”

  1. Comment: “……factors of TTS (older age, female gender), inflammaging-related pathways were identified to……” should be “……factors of TTS (older age, female gender), inflammation-related pathways were identified to……”.

Response: “inflammaging” is a key concept of this study. Thus, we consider that the term of “inflammaging” should not be changed.

Reviewer 3 Report

Blood biomarkers in Takotsubo syndrome point to an emerging role for inflammaging in endothelial pathophysiology.

it is a well-written article describing the scientific background, methodologies, supportive results, and discussion. A few suggestions to improve some existing issues:

Line 104 please specify “protocol 01-3-3”

Line 143 please add concentrations if known

Line 147 change 0,4 with 0.4, check also through the entire paper for similar mistakes i.e. table 2  

Line 397 uniform the font of the table 3 caption

Figure 4 WB image, i would improve the image or at least provide an image with a better contrast

Figure 5-6-7 could you provide a better resolution

Line 628 “new technologies like big-data analysis” please change with i.e. high throughput or omics techniques

Author Response

Ms. biomolecules-2419031: "Blood biomarkers in Takotsubo syndrome point to an emerging role for inflammaging in endothelial pathophysiology"

Response to the Reviewer 3

We are very grateful for your comments, which have substantially improved our manuscript. We considered the following points and summarized our responses.

Major comments

  1. Comment: Blood biomarkers in Takotsubo syndrome point to an emerging role for inflammaging in endothelial pathophysiology. It is a well-written article describing the scientific background, methodologies, supportive results, and discussion. A few suggestions to improve some existing issues:

Response: Many thanks for the important critiques which will help to improve over the quality of the manuscript. We added the requested information as detailed below.

  1. Comment: Line 104 please specify “protocol 01-3-3”

Response: This is the serial number in the the Hiroshima City Asa Hospital Research Ethics Committee registry (01-3-3) as previously described (Thal, S.C. et al. (2022) Hemorrhagic cerebral insults and secondary takotsubo syndrome: Findings in a novel in-vitro model using human blood samples. Int J Mol Sci 2022 Sep 30;23(19):11557. doi: 10.3390/ijms231911557.) We added this information.

  1. Comment: Line 143 please add concentrations if known.

Response: We apologize but unfortunately, we do not know the exact concentrations and they are not specified on the manufacturer's website.

  1. Comment: Line 147 change 0,4 with 0.4, check also through the entire paper for similar mistakes i.e. table 2.

Response: We apologize for this incorrectness. This is the German spelling. We checked the manuscript carefully. Table 2 was corrected throughout.

  1. Comment: Line 397 uniform the font of the table 3 caption.

Response: We apologize for this incorrectness. The caption was uniformed to Palatino 9 pt.

  1. Comment: Figure 4 WB image, i would improve the image or at least provide an image with a better contrast.

Response: We apologize but this is unfortunately not possible. In frame of this small cohort study, we had used all the material for the present study und 2 previous studies, ie Thal, S.C., Smetak:, M., Hayashi, K., Förster, C.Y. (2022) Hemorrhagic cerebral insults and secondary takotsubo syndrome: Findings in a novel in-vitro model using human blood samples. Int J Mol Sci 2022 Sep 30;23(19):11557. doi: 10.3390/ijms231911557.

And, Karnati, S., Rajenderan, R., Shityakov, S., Güntas, G., Liebisch, G., Kosanovic, D., Ergün, S., Nagai, M., Förster, C.Y. (2021) Lipid profiling of Takotsubo syndrome patient blood – does a disturbed equilibrium of lipid accumulation and immune response contribute to disease predisposition? Frontiers Cardiovasc. Med., 19 April 2022 | https://doi.org/10.3389/fcvm.2022.797154. There is no material left to redo this western blot, unfortunately.

  1. Comment: Line 628 “new technologies like big-data analysis” please change with i.e. high throughput or omics techniques

Response: We apologize for inappropriate wording. We changed to using high throughput or omics techniques for the analysis as requested by the reviewer.